# Potential Skin Health Benefits of Abalone By-Products Suggested by Their Effects on MAPKS and PI3K/AKT/NF-kB Signaling Pathways in HDF and HaCaT Cells

**DOI:** 10.3390/foods13182902

**Published:** 2024-09-13

**Authors:** Eun-A Kim, Nalae Kang, Jun-Ho Heo, Areumi Park, Seong-Yeong Heo, Chang-Ik Ko, Yong-Seok Ahn, Ginnae Ahn, Soo-Jin Heo

**Affiliations:** 1Jeju Bio Research Center, Korea Institute of Ocean Science and Technology (KIOST), Jeju 63349, Republic of Korea; euna0718@kiost.ac.kr (E.-A.K.); nalae1207@kiost.ac.kr (N.K.); unknown0713@kiost.ac.kr (J.-H.H.); areumi1001@kiost.ac.kr (A.P.); syheo@kiost.ac.kr (S.-Y.H.); 2Choung Ryong Fisheries Co., Ltd., Jeju 63612, Republic of Korea; rhckddlr01@naver.com (C.-I.K.); ecoil@hanmail.net (Y.-S.A.); 3Department of Food Technology and Nutrition, Chonnam National University, Yeosu 59626, Republic of Korea; gnahn@jnu.ac.kr; 4Department of Marine Biology & Convergence Engineering (Marine Biotechnology), University of Science and Technology (UST), Daejeon 34113, Republic of Korea

**Keywords:** abalone, viscera, by-product, skin moisturizing, HDF cells, HaCaT cells

## Abstract

Abalone, a marine edible gastropod with nutritional value, is a popular seafood delicacy worldwide, especially in Asia; however, viscera by-products are generally discarded during processing. Therefore, we investigated the skin health benefits of abalone viscera ultrasonic extract (AVU) in human dermal fibroblasts (HDFs) and human keratinocyte (HaCaT) cells. AVU showed valuable protein contents, indicating that it is a worthy and safe material for industrial application. AVU increased collagen synthesis production and messenger RNA (mRNA) expression of Collagen Type I Alpha 1, 2, and 3 chains through the transforming growth factor beta/suppressor of mother against the decapentaplegic pathway in HDF cells. AVU also increased hyaluronic acid production, upregulated Hyaluronan Synthases 1, 2, and 3, filaggrin and aquaporin3 mRNA levels, and downregulated hyaluronidase mRNA levels in HaCaT cells. Furthermore, mechanistic studies showed that AVU increased the phosphorylation of extracellular signal-regulated kinase, p38, and cyclic AMP response-binding protein activation. AVU activated the transcription factors, phosphoinositide 3-kinase, protein kinase B, and nuclear factor kappa B cell p65 and downregulated the degranulation of inhibitory kappa B in HaCaT cells. Studies of hyaluronic acid production in AVU by inhibiting EKR, p38 and NF-κB have shown that p38 MAPK and NF-κB signaling are pivotal mechanisms, particularly in the AVU. These results demonstrated that AVU produced from by-products may improve skin health and may thus be used as a functional food and cosmetics ingredient.

## 1. Introduction

Abalone, a marine edible gastropod, is a popular food resource due to its nutritional and pharmaceutical value. It is highly valued as a seafood delicacy in many parts of the world, especially Asia [1,2,3]. Therefore, abalone aquaculture has increased globally, reaching 190,000 tons yearly in 2019 [2]. Abalone viscera by-products, which accounts for approximately 15–25% of total weight, are normally discarded, contributing to environmental pollution and waste of potentially economically valuable resources [2,3,4]. However, the organic residues of these by-products are a rich source of proteins, lipids, and polysaccharides with various biological activities, such as anti-cancer, antioxidant, anti-inflammatory and immune-enhancing activities [3,5,6]. Notably, abalone viscera has antioxidant, anti-inflammatory, anti-cancer, antimicrobial, and anti-hypertensive activities [3,6,7]. Therefore, since the fishing industry now produces enormous amounts of organic residues as industrial waste detrimental to the environment and human and animal health, various effective ways of using these residues are being explored [2,6].

Food and allied industries require safe and economically efficient processing methods for producing functional materials. Among the various extraction methods, ultrasonication is one of the rapidly emerging techniques devised to minimize processing, enhance quality, and ensure product safety [8,9]. This method is referred to as a green technology owing to the reduced use of organic solvents and processing time, better quality, energy efficiency, and the inactivation of certain microorganisms [8,10]. Therefore, the food and allied industries are using this effective extraction method to extract and analyze various components such as proteins, flavonoids, pigments, and minerals from a variety of matrices, including seafood, vegetables, and fruits [10,11,12].

The skin, which includes the epidermis, dermis, and subcutaneous layers, is one of the largest and most important organs of the human body. It involves a complex biological process influenced by endogenous (cellular metabolism, hormone, genetics) and exogenous (pollution, chemicals, UV radiation) factors [13,14,15]. Therefore, functional and medicinal substances are administered orally to create healthy skin, and cosmetics that beautify the exterior are used [13]. Consequently, many studies have attempted to find novel substances that can positively impact skin health [13]. Skin hydration is crucial to having healthy skin because proper skin hydration increases skin flexibility, protects it from damage, and enables a process of desquamation by providing an environment in which hydrolytic enzymes can be activated [16]. Collagen, one of the most ubiquitous proteins in the human body, is a major component of the extracellular matrix (ECM). Type 1 and 3 collagens, the most abundant ECM component, account for approximately 95% of all known collagen types and comprise broad extracellular fiber in the dermis of human adult skin [17,18,19]. Collagen types 1 and type 3 account for approximately 85–95% and 8–11% of the total collagen on human adult skin [18,19]. The transforming growth factor (TGF)-beta, a regulator of ECM synthesis, stimulates the proliferation and activation of dermal fibroblasts during the suppressor of mother against the decapentaplegic (SMAD) 2/3 pathway [17,19]. Filaggrin, the main precursor protein of the amino acid-derived components of the natural moisturizing factor (NMF), is a structural protein required for fully competent epidermal barrier function [20].

Furthermore, hyaluronic acid (HA) and aquaporins (AQP3), a critical ECM molecule that maintains the skin’s moisture content, capture large amounts of water, and are responsible for preventing moisture loss from the epidermis [14,21]. There are three primary forms of HA synthases (HAS1, HAS2, and HAS3), and each has a different enzyme activity and enables the synthesis of different HA lengths [21]. Furthermore, among the 13 AQPs in mammals, APQ3 is a membrane protein that acts as a channel that can transport glycerol and water [21,22]. HAS2 is mainly expressed in normal human cells. It maintains skin moisture and homeostasis and reduces intrinsically aged human skin by upregulating HAS2 through the mitogen-activated protein kinase (MAPK) and phosphoinositide 3-kinase/protein kinase B (PI3K/Akt) signaling pathways [16,22].

Therefore, we aimed to produce a safe and useful extract from abalone viscera, a by-product, using ultrasonic extraction and utilize it as a valuable industrial substance. We also identified abalone visceral ultrasonic extraction (AVU) components and the molecular mechanisms of its moisturizing effects that may be beneficial to skin health in human dermal fibroblasts (HDFs) and keratinocyte cells.

## 2. Materials and Methods

### 2.1. AVU Preparation

#### 2.1.1. AVU Extraction

Abalone was obtained from the fishing village market on Jeju Island, Republic of Korea. After removing the shells, the muscle and visceral tissues were separated and washed under tap water. The visceral tissue (1 kg) was dissolved in distilled water using an ultrasonicator (vibration: 1200 W; stirring speed: 60 rpm; temperature: 4–8 °C) for 4 h. The extract was centrifugated at 4500 rpm for 10 min and filtered through filter paper at 1 µm and 0.45 µm, respectively. The supernatant obtained was freeze-dried to obtain the AVU.

#### 2.1.2. Analysis of the Chemical Composition of AVU

AVU yields were calculated by subtracting the dried weight of the residue from 1 mL of AVU and were expressed as a percentage. The protein content was analyzed using the bicinchoninic acid (BCA) method, with bovine serum albumin as a reference standard. The total polysaccharide content was analyzed using the phenol–sulfuric acid method with glucose as the reference standard, and the total polyphenolic content was analyzed with gallic acid as a reference standard by modifying the methods described by Fernando et al. (2017) [23].

#### 2.1.3. Analysis of Microorganisms

The microbial analysis of AVU was performed using the total bacteria count, coliform bacteria count, total fungal count, and *Staphylococcus aureus* quantification using the standard plate count, dry rehydratable film, potato dextrose agar media, and Baird–Parker agar media methods following the standard method of the Korean Food Code [24].

#### 2.1.4. Profile of Amino Acid Composition

The amino acid composition was analyzed using a previously developed high-performance liquid chromatography (HPLC) method. The AVU mixture was added to 75% ethanol, extracted by ultrasonication for 1 h, and then incubated for 24 h at room temperature. The mixture was filtered with a 0.2 µm syringe filter and used for the HPLC sample. The HPLC system (Dionex Ultimate 3000, Thermo, Waltham, MA, USA) had fluorescence and ultraviolet ray detectors. Amino acids were determined using a VDSpher 100 C18-E column. The experiment conditions were as follows: the mobile phase (A) was 40 mM of sodium phosphate buffer (pH = 7) and (B) was 3DW:acetonitrile:methanol (10:45:45), where injection volume was 0.5 µL, and the column temperature was 40 °C.

### 2.2. In Vitro Skin Health Effects

#### 2.2.1. Cell Culture

HDF and human keratinocyte (HaCaT) cell lines were purchased from the American Type Culture Collection (ATCC, Manassas, VA, USA). HDF and HaCaT cells that had been cultured for 5–9 and 35–45 passages, respectively, were used for the experiments. Dulbecco’s Modified Eagle Medium (DMEM)/Nutrient Mixture F-12 (DMEM/F-12) mixed at a ratio of 3:1 was supplemented with 10% fetal bovine serum (FBS), and a 1% penicillin and streptomycin mixture to culture the HDF cells. Furthermore, DMEM media supplemented with 10% FBS and a 1% penicillin and streptomycin mixture were also used to culture the HaCaT cells. These cells were incubated at 37 °C with a 5% carbon dioxide humidified atmosphere.

#### 2.2.2. Determinations of Procollagen Type 1 C-Peptide and Hyaluronan Levels Using an Enzyme-Linked Immunosorbent Assay

HDF or HaCaT cells were seeded at the density of 2 × 10^4^ or 2 × 10^5^ cells/well into 96- or 24-well plates, respectively. After incubating for 16 h, the supernatant was removed, the cells were washed twice using DPBS, and serum-free media were added. The cells were then treated with various concentrations of AVU (50, 100, and 200 µg/mL), TGFβ (10 ng/mL), and retinoic acid (10 µM) for 24 h. The procollagen type 1 production was evaluated using the Procollagen Type I C-peptide (PIP) kit assay (Takara Bio Inc. Shiga, Japan), and the HA production was measured using the hyaluronan kit assay (R&D System, Minneapolis, MN, USA), based on the manufacturer’s instruction. HaCaT cells were pretreated with an ERK inhibitor (PD; PD98059), p38 inhibitor (SB; SB203580), and NF-kB inhibitor (PDTC; ammonium pyrrolidinedi-thiocarbamate) followed by treatment with AVU (200 µg/mL) to assess HA production using the hyaluronan kit assay. 

#### 2.2.3. Reverse Transcription Polymerase Chain Reaction Analysis

Reverse-transcription polymerase chain reaction (RT-PCR) analysis was performed following the methods described by Kang et al. [25]. Total RNA was isolated using a TRIzol reagent following the manufacturer’s instructions. The complementary DNA was synthesized from 1 µg of the total RNA using the Prime Script RT Reagent Kit. The RT-PCR analysis was performed using a Quant Studio 3 real-time PCR system (Applied Biosystems, Thermo, Waltham, MA, USA) with a modified method and reagents. The sequences of primer sets used included the following: COL1A1 sense, 5′-AGCCCTGGTGAAAATGGAGC-3′ and antisense, 5′-TCATTTCCACGAGCACCAGC-3′; COL1A2 sense, 5′-GGCCCTCAAGGTTTCCAAGG-3′ and antisense, 5′-CACCCTGTGGTCCAACAACTC-3′; COL3A1 sense, 5′-TTGAAGGAGGATGTTCCCATCT-3′ and antisense, 5′-ACAGACACATATTTGGCATGGTT-3′; HAS1 sense, 5′-CCACCCAGTACAGCGTCAAC-3′ and antisense, 5′-CATGGTGCTTCTGTCGCTCT-3′; HAS2 sense, 5′-GTCGAGTTTACTTCCCGCCA-3′ and antisense, 5′-ATCACACCACCCAGGAGGAT-3′, HAS3 sense, 5′-GATTTCCTTCCTGAGCAGCG-3′ and antisense, 5′-TGTTGCGGTACATGCCCAAG-3′. Hyaluronidase 1 sense, 5′-GATGGCTGTGGAGTTCAAATG-3′ and antisense, 5′-CCCAGAGTGCATTAGGTTCTC-3′, Filaggrin sense, 5′-GGCTAAGTGAAAGACTTGAAGAGA-3′ and antisense, 5′-AATAGACTATCAGTGGTGTCATAGG-3′, AQP3 sense, 5′-TGCAATCTGGCACTTCGC-3′ and antisense, 5′-GCCAGCACACACACGATAA-3′, and β-actin sense, 5′-CACTGTGCCCATCTACG-3′ and antisense, 5′-CTTAATGTCACGCACGATTTC-3′.

#### 2.2.4. Western Blotting

The effects of AVU on the protein expression levels of HAS2, AQP3, phospho (p)-extracellular signal-regulated kinase (ERK), ERK, p-c-Jun N-terminal kinase (JNK), JNK, p-p38, p38, p-cyclic AMP response-binding protein (CREB), CREB, p-PI3K, PI3K, p-AKT, AKT, inhibitory kappa B (IkB), p-p65, p-65, and β-actin were analyzed through Western blotting. The HaCaT cells were seeded as explained above and treated with AVU for 2 h or 24 h incubation periods following the method of Kim et al. [6]. The protein concentrations of cell lysates were measured using a BCA™ protein assay kit following the manufacturer’s instructions. The primary (1:1000 dilution, Cell Signaling Technology, Beverly, MA, USA or Santa Cruz Biotechnology, Santa Cruz, CA, USA) and secondary antibodies (anti-mouse IgG and anti-rabbit IgG, 1:3000 dilution, Santa Cruz Biotechnology) were analyzed. The protein bands were detected using FUSION SOLO (Vilber Lourmat, Marne-la-Vallée, France), and the intensity quantification of Western blot results was completed with ImageJ software 1.46r. 

### 2.3. Statistical Analysis

All data were recorded in triplicate and reported as means ± standard deviation. Statistical analysis for comparing the data was implemented using GraphPad Prism 10 software (GraphPad Software, San Diego, CA, USA) using a one-way analysis of variance followed by Dunnett’s post hoc test. Statistical significance was set at *p* < 0.5.

## 3. Results

### 3.1. Chemical Composition and Microbiological Testing of AVU

Table 1 shows the chemical composition of AVU. The AVU yield was 6.53%, and the protein, polysaccharide, and total phenolic contents were 39.01%, 8.63%, and 1.48%, respectively. These results showed the rich protein contents of AVU. For essential amino acid analysis, AVU showed especially high contents of alanine (65%), arginine (7%), and glycine (6%) (Table 1). Based on microbiological testing, the number of bacteria was expressed as colony-forming units (CRU/mL). The total bacteria and fungal counts were below the detection limit in AVU (0/mL). Furthermore, coliform bacteria and *Staphylococcus aureus* were not observed in AVU (0/mL; Table 1). Therefore, AVU functionality for increased usability for functional foods or cosmetics was confirmed.

### 3.2. Collagen Synthesis Effects of AVU in HDF Cells

The cytotoxicity of AVU (50, 100, and 200 µg/mL) on HDF cells was measured using the 3-[4,5-dimethylthiazol-2-yl]-2,5 diphenyl tetrazolium bromide (MTT) assay (Figure 1a). AUV was confirmed to have a cell proliferation effect by increasing cell viability in a concentration-dependent manner ranging from 50 to 200 µg/mL. Notably, further experiments have used these concentrations based on the cell viability data from AVU. Collagen was synthesized from procollagen, a precursor molecule containing additional peptide sequences [26]. We utilized the PIP kit to determine collagen synthesis production. In Figure 1b, the collagen level was significantly increased at 50–200 µg/mL of AVU. Therefore, we analyzed the transcription levels of Collagen Type I Alpha 1 chain (COL1A1), COL1A2, and COL3A1 using RT-PCR to confirm the synthesis of the type 1 and 3 collagens, the major isotypes produced by fibroblasts (Figure 1c–e). The results revealed increased COL1A1 and COL1A2 messenger RNA (mRNA) transcript levels at 200 µg/mL of AVU. However, the COL3A1 mRNA level significantly increased at 100–200 µg/mL of AVU. TGF-β regulates collagen homeostasis by collagen production and degradation through the SMAD pathway [27]. Figure 2 shows that the treatment of AVU at 50–200 µg/mL markedly upregulated TGF-β compared with the control, and the phosphorylation of SMAD2 expression was also significantly increased. Therefore, our findings showed that AVU upregulates collagen synthesis through the TGF-β/SMAD pathway in HDF cells. 

### 3.3. Moisturizing Effect of AVU in HaCaT Cells

To determine cytotoxicity, the MTT assay was used to assess the cell viability of HaCaT cells treated with different AVU concentrations (50, 100, and 200 µg/mL). AVU at various concentrations significantly increased the cell viability of HaCaT cells (Figure 3a). Therefore, AVU showed cell proliferation effects without cytotoxicity at 50–200 µg/mL. HA, known as hyaluronan, has more skin regeneration, with moisturizing and anti-aging effects than other substances [28]. In Figure 3b, AVU increased the HA content in a concentration-dependent manner. The HA levels with 100 and 200 µg/mL AVU were 108.8% and 124.6% compared with those in control groups, respectively. Hyaluronidase is an enzyme that degrades HA, and HAS enzymes are the membrane-associated enzymes that synthesize HA [22]. AVU decreased the hyaluronidase mRNA levels and increased the HAS1, 2, and 3 mRNA levels (Figure 3c–f). In addition, hydrolytic enzymes decompose filaggrin in the stratum corneum into free amino acids to form the NMF, which regulates moisture and the skin barrier [29]. AVU treatment significantly increased filaggrin transcripts in a concentration-dependent manner (Figure 3g). Furthermore, AQP3, another factor associated with skin moisturization, significantly increased mRNA levels (Figure 3h). HAS2 and AQP3, in particular, help maintain the skin moisture content, and AVU demonstrated a concentration-dependent increase in HAS2 and AQP3 protein levels as confirmed by Western blotting (Figure 3i). Consequently, AVU has been observed to upregulate the levels of HAS1, HAS2, HAS3, filaggrin, and AQP3 mRNA, downregulate the hyaluronidase mRNA levels, and increase HAS2 and AQP3 protein levels in HaCaT cells, indicating that AVU exerts a moisturizing effect by increasing HA production. 

### 3.4. Moisturizing Effect of AVU through the MAPK and PI3K/Akt/NF-kB Signaling Pathway in HaCaT Cells

The MAPKS/CREB pathway is responsible for the synthesis of HA, and the PI3K/Akt/nuclear factor kappa B (NF-kB) pathway also acts as a HAS expression mechanism in human skin cells [16,30]. Therefore, we determined whether this signaling is associated with MAPKS/CREB and PI3K/Akt/NF-kB pathways through Western blotting. Figure 4a shows that AVU treatment significantly increased the phosphorylation of ERK, p-38, and CREB in HaCaT cells compared with the control groups. AVU activated the transcription factors, p-PI3K, p-Akt, and p-NF-kB p65 in HaCaT cells at 50, 100, and 200 µg/mL concentrations in a concentration-dependent manner and suppressed the degranulation of IkB (Figure 4b). We confirmed the protein phosphorylation when AVU was treated with inhibitors such as ERK (PD; PD98059), p38 (SB; SB203580), and NF-kB (PDTC; ammonium pyrrolidinedithiocarbamate). The treatment with inhibitors such as PD, SB, and PDTC showed decreased p-ERK, p-p38, and p-p65 protein levels compared with the control group and the AVU group. Furthermore, the co-treatment with AVU and inhibitors groups resulted in a significant reduction in the phosphorylation protein levels compared with the AVU group. Therefore, we confirmed that AVU increased HA by regulating the ERK and p38 MAPKs/CREB and PI3K/Akt/NF-kB signaling pathways in HaCaT cells.

### 3.5. Role of Hyaluronic Acid Production in AVU-Induced EKR, p38 and NF-kB Activation in HaCaT Cells

To determine the effects of AVU on cell viability and hyaluronic acid production with ERK, p38, and p65 pathways, we performed studies on HaCaT cells using inhibitors such as PD, SB, and PDTC. As shown in Figure 5a, the co-treatment with AVU and inhibitor groups and the inhibitor groups showed no toxicity in HaCaT cells compared with the control group as confirmed by the MTT assay. In Figure 5b, in the inhibitor groups, such as those treated with PD, SB, and PDTC, a notable decline in HA production compared with the AVU group was reported. And the co-treatment with AVU and inhibitor groups also decreased HA production compared with the AVU group. In particular, SB and PDTC exhibited the strongest inhibitory effect on AVU-mediated hyaluronic acid production.

These results suggest that AVU may enhance HA production by regulating the MAPKs/CREB and PI3K/Akt/NF-kB pathways. Especially, AVU represents a pivotal mechanism through which p38 MAPK and NF-κB signaling contributes to the observed benefits for skin health.

## 4. Discussion

Abalone is a valuable gastropod. The muscles are used for food. However, the viscera constitutes over 20% of the total weight and is generally considered waste in the food industry. Therefore, in this study, we confirmed the skin health effect of AVU obtained from marine by-products in HDF and HaCaT cells.

Ultrasonication technology is environmentally friendly and efficiently extracts bioactive compounds such as proteins from plants, animal tissues, and other natural materials [10,11,12,31,32]. Therefore, ultrasonic extraction is utilized in the industry as an efficient water-based method. After this environmentally friendly extraction, AVU showed a lower yield, but high protein content and below the detection limit of microorganisms (0/mL), suggesting its suitability for functional food or cosmetics production.

The NMF is a combination of free amino acids (40%), pyrrolidone carboxylic acid (12%), lactate, sugars, and urea [33]. Among the free amino acids that constitute the NMF, serine is the most abundant, followed by glycine, alanine, histidine, ornithine, citrulline, and arginine within the NMF [34]. AVU, which has a sufficient amount of protein, contained high contents of alanine (65.3%), arginine (7.3%), and glycine (6.5%), accounting for 78.7%. Notably, alanine is known to balance moisture levels on the skin, thus providing improved hydration [35]. Therefore, we hypothesized that AVU was an ingredient that could effectively moisturize the skin. Consequently, we confirmed the moisturizing effect of AVU in HDF and HaCaT cells.

Collagen, a structural protein, constitutes a large part of the connective tissue, especially bones, tendons, joints, and skin [36,37]. In skin, collagen provides a support matrix underpinning healthy skin and is a critical determinant in preserving skin firmness and elasticity [37]. Among the five types of collagen, the main collagens, types 1 and 3, account for 95% of skin collagen [17,37]. AVU induced the production of procollagen type 1 and upregulated the mRNA expression of COL1A1, COL1A2, and COL3A1 by increasing TGFβ protein and the phosphorylation of SMAD 2 protein expression. TGFβ crucially enhances the expression of several collagens, such as types 1 and 3 [17]. In a previous study, abalone viscera ethanol extract showed an effect on procollagen type 1 synthesis of approximately 705 ng/mL at 10–12.5 µg/mL; however, the extract began to decrease at 25 µg/mL and further reduced at 200 µg/mL compared with that in control groups. Therefore, the procollagen type 1 synthesis effects were approximately two-fold compared with that in the control group at 12.5 µg/mL [38]. In contrast, the synthesis activity of AVU was 545.8, 762.7, and 1066.8 ng/mL at 50–200 µg/mL, respectively. High concentrations showed eight-fold procollagen type 1 synthesis effects compared with that in the control group. HA is one of the most widely used active components in cosmetic formulation because it improves skin moisture content and reduces aging symptoms and signs [28]. It is synthesized by HAS1, 2, and 3. HAS1 exhibits the highest enzymatic activity, followed by HAS2 and HAS3 [21,39]. AQP3 is also crucial in maintaining skin moisture. It is one of the moisture channel-specific proteins composed of 13 families (AQP0–12) that transports moisture from the dermis to the epidermis [22,40]. Filaggrin protein is the main source of several major NMF components in the stratum corneum [20]. The moisturizing effect of AVU was investigated based on the expression of HA, AQP, and filaggrin. AVU enhanced these factors. Therefore, AVU is indicated to maintain moist skin by regulating NMFs such as HA, AQP, and filaggrin. AVU also reduced hyaluronidase contents, leading to a loss of strength and moisture and ultimately causing skin aging [41]. AVU accelerates HA production by elevating the expression of HAS1, 2, and 3, thereby simultaneously enhancing skin hydration. Furthermore, we confirmed how AVU regulates the expression of these moisturizing factors. HAS2, the most strongly expressed isoform in keratinocytes, helps retain skin moisture and maintains homeostasis by regulating HA synthesis [22,39]. Therefore, we attempted to identify the upstream signal triggered by AVU in HAS2 upregulation. HAS2 regulates the PI3K/AKT and MAPK signaling pathways for promoting NF-kB and CREB-binding regions [16,22]. The MAPK signaling pathway promotes the expression of HAS and AQP3, producing a moisturizing effect [14]. AVU showed increased phosphorylation of ERK, p38, and CREB. Furthermore, AVU increased the phosphorylation of PI3K, AKT, and p-65 and decreased IkB. Therefore, we unveiled the interactions between AVU and the signaling pathway using inhibitors. The inhibitor groups such as PD (ERK inhibitor), SB (p38 inhibitor), and PDTC (p65 inhibitor) showed a reduction in the phosphorylation of ERK, p38, and p65 protein. The treatment of AVU with inhibitors groups also indicated decreased phosphorylation levels of these proteins compared with the AVU group. In addition, the inhibitor groups and the co-treatment with AVU and inhibitor groups indicated significantly decreased hyaluronic acid production compared with the AVU group. Especially, SB and PDTC inhibited AVU-mediated hyaluronic acid production. AVU is thought to be a key mechanism by which p38 MAPK and NF-κB signaling is responsible for the observed skin health benefits.

In conclusion, AVU, as a by-product from fishery manufacturing, offers promise through its highly valuable utilization and application as a potential active substance that may help promote skin health through cosmetics and functional foods.

## Figures and Tables

**Figure 1 foods-13-02902-f001:**
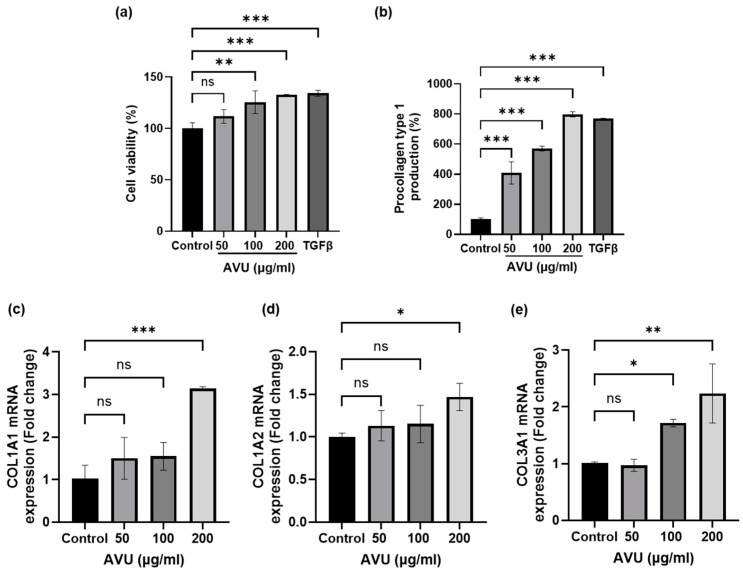
Collagen synthesis effects of AVU in HDF cells. Cells were treated with different AVU concentrations (50, 100, and 200 µg/mL) and TGFβ (10 ng/mL) for 24 h. (**a**) Cell viability was measured using the MTT assay. (**b**) The supernatants were collected and measured for procollagen type 1 production using the Procollagen Type 1 C-peptide kit. (**c**–**e**) Cell lysates were extracted, and mRNA levels of COL1A1, COL1A2, and COL3A1 were analyzed using real-time PCR. Values are expressed as mean ± SD of triplicate experiments. ns: not significant, * *p* < 0.05, ** *p* < 0.01; *** *p* < 0.001 indicate significant differences from the control groups.

**Figure 2 foods-13-02902-f002:**
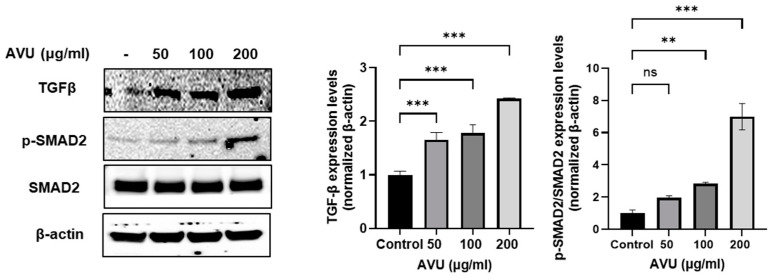
Induction of collagen synthesis effects of AVU through the TGFβ/SMAD2 signaling pathway in HDF cells. Cells were treated with different AVU concentrations (50, 100, and 200 µg/mL) for 24 h. Cell lysates were extracted, and protein levels of TGFβ, p-SMAD2, SMAD2, and β-actin were analyzed using Western blot. Quantitative fluorescence analysis was performed using the ImageJ software. Values are expressed as mean ± SD of triplicate experiments. ns: not significant, ** *p* < 0.01; *** *p* < 0.001 indicate significant differences from the control groups.

**Figure 3 foods-13-02902-f003:**
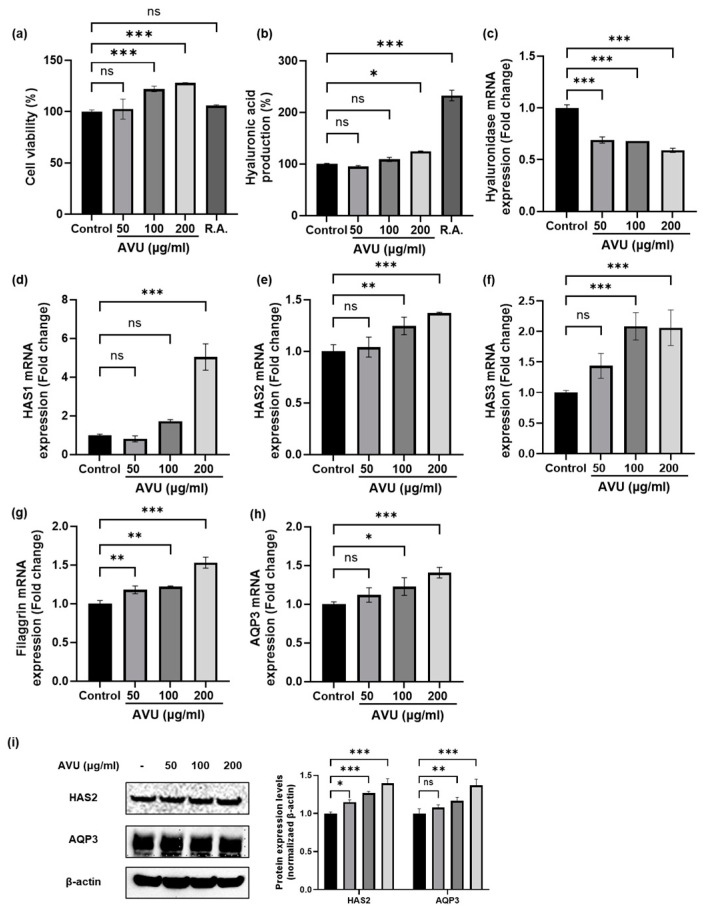
Moisturizing effect of AVU in HaCaT cells. Cells were treated with different AVU concentrations (50, 100, and 200 µg/mL) and retinoic acid (10 µM) for 24 h. (**a**) Cell viability was measured using the MTT assay. (**b**) The supernatants were collected and measured for hyaluronic acid production using the hyaluronan kit. (**c**–**h**) Cell lysates were extracted, and mRNA levels of hyaluronidase, HAS1, HAS2, HAS3, filaggrin, and Aquaporin3 were analyzed using real-time PCR. (**i**) HAS2 and AQP3 protein levels were assessed using Western blotting, and quantitative fluorescence analysis was performed using ImageJ software. Values are expressed as mean ± SD. of triplicate experiments. ns: not significant, * *p* < 0.05, ** *p* < 0.01; *** *p* < 0.001 indicate significant differences from the control groups.

**Figure 4 foods-13-02902-f004:**
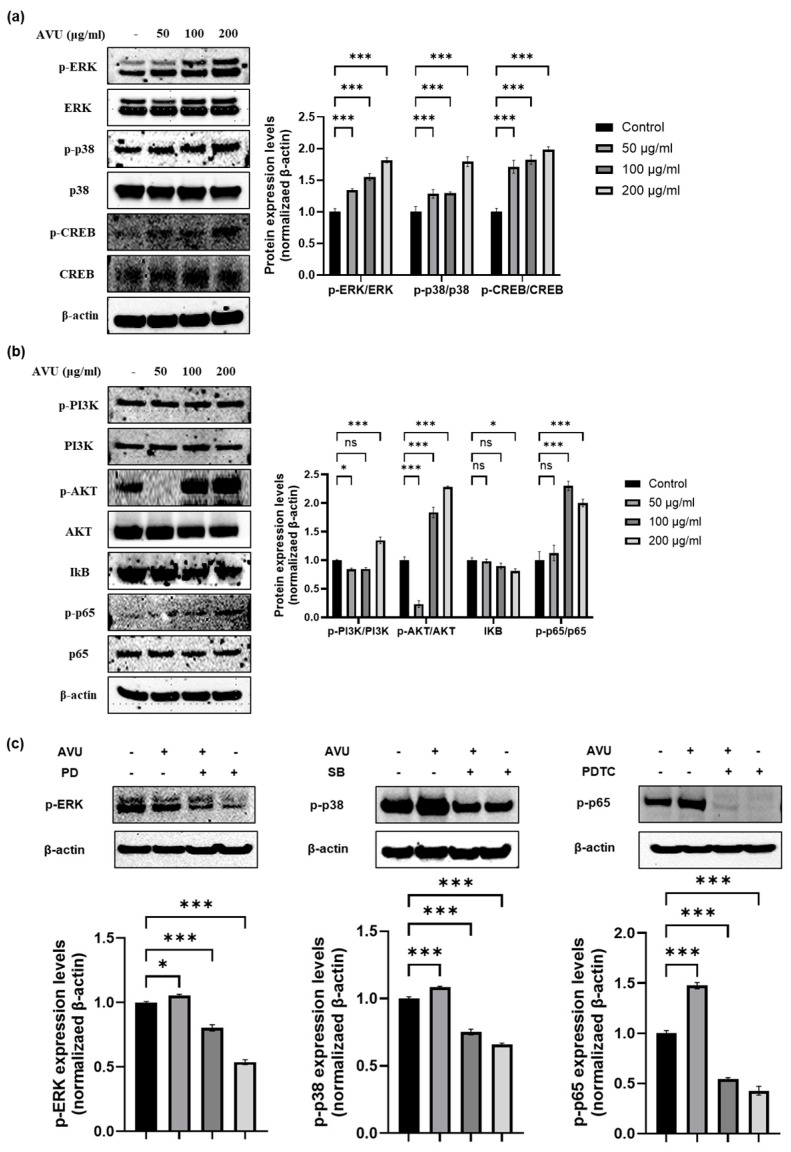
Regulation of AVU moisturizing factors through the MAPK/CREB and PI3K/AKT/NF-kB pathways in HaCaT cells. Cells were treated with different AVU concentrations (50, 100, and 200 µg/mL) for 2 h. (**a**,**b**) Cell lysates were extracted, and protein levels of p-ERK, ERK, p-p38, p38, p-CREB, CREB, p-PI3K, PI3K, p-AKT, AKT, IkB, p-p65, p65, β-actin were analyzed using Western blot. (**c**) Cells were treated with AVU (200 µg/mL) and ERK inhibitor (PD; PD98059), p38 inhibitor (SB; SB203580), and NF-kB inhibitor (PDTC; ammonium pyrrolidinedi-thiocarbamate) and inhibitor pretreatment was followed by AVU to assess protein levels using Western blotting. Quantitative fluorescence analysis was performed using the ImageJ software. Values are expressed as mean ± SD of triplicate experiments. ns: not significant, * *p* < 0.05, *** *p* < 0.001 indicate significant differences from the control group.

**Figure 5 foods-13-02902-f005:**
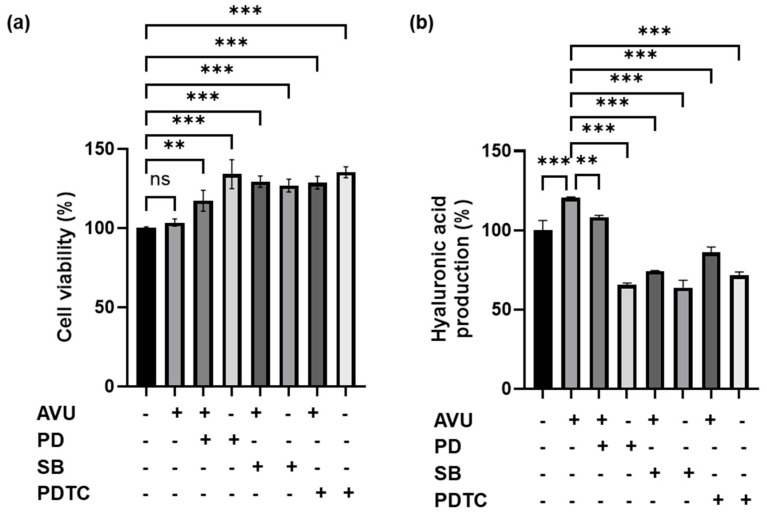
Cell viability and hyaluronic acid production effect of AVU and co-treatment with AVU and inhibitors. (**a**) Cell viability of HaCaT cells was measured using the MTT assay. (**b**) The HaCaT cells supernatants were collected and measured for hyaluronic acid production using the hyaluronan kit. Values are expressed as mean ± SD of triplicate experiments. ns: not significant, ** *p* < 0.01; *** *p* < 0.001 indicate significant differences from the control group or the AVU group. PD (PD98059), SB (SB203580), and PDTC (ammonium pyrrolidinedi-thiocarbamate).

**Table 1 foods-13-02902-t001:** Chemical composition and microbiological analysis.

Proximate Composition (%)	
Yield	6.53 ± 0.25
Protein contents	39.01 ± 1.76
Polysaccharide contents	8.63 ± 0.22
Total phenolic contents	1.48 ± 0.08
Microorganisms (/mL)	
Total bacteria count	0
Coliform bacteria	0
Total fungal count	0
*Staphylococcus aureus*	0
Amino acid composition (%)	
Aspartic acid	2.80
Glutamic acid	5.82
Serine	2.00
Glutamine	1.99
Histidine	0.44
Glycine	6.07
Threonine	1.63
Arginine	7.33
Alanine	65.30
Tyrosine	0.68
Valine	1.60
Isoleucine	0.99
Lysine	2.38
Proline	0.97

## Data Availability

The original contributions presented in the study are included in the article, further inquiries can be directed to the corresponding author.

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
