# Peer review of "Potential Skin Health Benefits of Abalone By-Products Suggested by Their Effects on MAPKS and PI3K/AKT/NF-kB Signaling Pathways in HDF and HaCaT Cells"

_foods, 2024, doi:10.3390/foods13182902_

Round 1

Reviewer 1 Report

Comments and Suggestions for Authors

This manuscript described the skin-moisturizing effects and molecular mechanisms of abalone by-products viscera extract. The results are some of interest, which would promote the development and application of abalone by-products as functional factors and cosmetics ingredient.

However, the low yield rete of the extract refer to underutilization. The ca. 6% extract rate means there are 94% compoments of abalone by-products viscera not used.

Why the authors extract with water in ultrasonicator, but not other extract approaches based the main compoments of the by-products, such as enzymatic hydrolysis? The authors should clarify and discuss this point in the manuscript.

Reviewer 2 Report

Comments and Suggestions for Authors

In this study, the authors examine the effects of an abalone viscera extract (AVU) on dermal fibroblasts and keratinocytes. Several issues are noted as stated below:

1. Title: The title should focus on 'skin health' or ECM deposition rather than 'skin hydration' which is difficult to determine from cultured monolayers of fibroblasts and keratinocytes. It is also recommended to use either "byproducts" or "viscera extract" instead of both to improve readability.

2. Abstract: The sentence "AVU did not detect microorganisms" should be deleted or rephrased.

3. Introduction: Some background should be provided regarding ultrasonication, its applications in preparing viscera extracts, and what is expected in the solution (e.g., proteins rather than fats etc.).

4. How was the AVU made free of microbes? As market bought abalone are not expected to be free of bacteria, this topic should be discussed, including the potential role of ultrasonication.

5. Cell culture: Please provide passage numbers of the cell lines used in this study, if available. Also, was AVU added to culture media containing FBS, or one without it? This is important as the constituents of FBS could potentially interact with those from AVU.

6. All findings from mRNA studies (e.g., those for HAS1 or AQP3) must be validated by protein assays (western blot, immunostaining or ELISA). Without evidence of protein synthesis, no biological significance could be determined.

7. The biggest flaw in this study is the lack of causality determined between the cell signaling pathways studied (e.g., ERK, Akt), and the outcomes measured (levels of collagen, AQP3 etc.). For example, to confirm that the collagen expression in response to AVU was caused by ERK (or p38), an experiments should have been performed with AVU and inhibitors of ERK (or p38). If the inhibition of ERK (or p38) by a specific chemical inhibitor blocked the AVU mediated rise in collagen synthesis, only then a causal relations hip to the signaling pathway (ERK or p38) would be ascertained. Such studies must be performed before making conclusions about the role/s of cell signaling pathways involved.

8. It would be interesting to know (or at least hypothesize) about how the components of AVU could activate cellular signaling pathways (specific receptors?) and the exact components which are involved in these processes.

Comments on the Quality of English Language

OK, but minor edits needed 

Reviewer 3 Report

Comments and Suggestions for Authors

The manuscript entitled “Skin-moisturizing effects of Abalone by-products viscera extract in HDF and HaCaT cells by regulating MAPKS and PI3K/AKT/NF-kB signaling pathways” reported the high value utilization of abalone viscera ultrasonic extract (AVU) in skin-moisturizing field. This work was valuable and may attract the readers and professors in skin protect field. However, some questions should be discussed.

1. Abalone viscera ultrasonic extract (AVU) whether can be identified as food ?

2. Skin-moisturizing effects was focus on the skin which lies in the interface between inner condition and outer environment of body. Thus, external use is not the application of traditional food.

3. The western results were blurry. Please upload original, unedited images for all blots used to generate the figures. These files should be the raw, uncropped scans or photographs of the entire membrane, labelled with molecular weight marker and the protein of interest identified.

4. The positive control group was miss in this manuscript file.

5. There are some careless mistakes in this peer preview file, which should be rechecked and revised.

Comments on the Quality of English Language

average level

Round 2

Reviewer 2 Report

Comments and Suggestions for Authors

A few comments regarding the revised manuscript:

1. Overall, most comments from this reviewer have been successfully addressed.

2. On line 277, what is being examined is protein phosphorylation rather than protein 'expression', as the MAPK inhibitors  block phosphorylation of the respective MAP kinases without changing their total protein levels. Similar changes should be made throughout the paper, especially in Figure 4.

3. Based on Figure 5 data, it is not convincing that PD, SB or PDTC has any effect on the AVU mediated increase in procollagen type I expression. Please address this and justify why these pathways (and not other signaling pathways were studied here).

4. Based on figure 5 data, it appears that only SB (but not PD or PDTC) inhibited AVU mediated hyaluronic acid production. This must be validated by showing a statistically significant decrease from the 'AVU alone' group. Interestingly enough, use of PD, SB and PDTC by itself (i.e., in absence of AVU) also decreased hyaluronic acid production/release below the baseline (no AVU, no treatment) group. The potential mechanisms and significance of this finding must be discussed.

Comments on the Quality of English Language

It is acceptable.
